# Impact of Type 2 Diabetes Mellitus on the Prognosis of Non-Small Cell Lung Cancer

**DOI:** 10.3390/jcm12010321

**Published:** 2022-12-31

**Authors:** You Lu, Yaohua Hu, Yi Zhao, Shuanshuan Xie, Changhui Wang

**Affiliations:** 1Department of Respiratory Medicine, Shanghai Tenth People’s Hospital, Tongji University School of Medicine, Shanghai 200072, China; 2Department of Pulmonary Medicine, Shanghai Chest Hospital, Shanghai Jiaotong University, Shanghai 200025, China; 3Department of Emergency, Shanghai Tenth People’s Hospital, Tongji University School of Medicine, Shanghai 200072, China

**Keywords:** non-small cell lung cancer, diabetes, prognosis, metformin

## Abstract

Objective: Type 2 diabetes mellitus (T2DM) is the most common metabolic disease and is characterized by sustained hyperglycemia. The impact of T2DM on the survival of lung cancer patients remains controversial. The aim of this study was to investigate the associations of type 2 diabetes with lung cancer mortality. Methods: From January 2019 to January 2020, 228 patients with non-small cell lung cancer (NSCLC) staging earlier than IIIA were included. Results: In our study, we found that the overall survival (OS) and progression-free survival (PFS) of lung cancer patients with diabetes was longer than non-diabetes group. Diagnosed T2DM was associated with the prognosis of lung cancer after adjusting for age and covariates. The association between T2DM and OS was influenced by age, stage of cancer and cancer treatment, as well as whether taking metformin was associated with the OS of lung cancer. However, with the adjustment for age and covariates, the relation trended to lose statistical significance. Conclusion: T2DM is an independent prognostic factor for patients with NSCLC staging before IIIA. The patients with both NSCLC and T2DM trended to having a longer OS, possibly due to metformin.

## 1. Introduction

Lung cancer is becoming one of the most common cancer-related causes of death globally [1,2]. According to estimates from GLOBOCAN 2020, all age-standardized incidence of lung cancer was 34.80/100,000 population, and lung cancer was the leading cause of cancer death with an age-standardized rate of cancer mortality of 30.20/100,000 population [3]. Approximately 85% of these lung cancer patients are diagnosed with non-small cell lung cancer (NSCLC). NSCLC is treated with surgery, radiation therapy, chemotherapy, specific and targeted therapy or a combination of these different approaches [4]. Over the past 20 years, the treatment of NSCLC has evolved from the empirical use of cytotoxic drugs to better tolerated and effective regiments by targeting specific molecular subtypes of the disease. Even if the progress has been promising with the advent of various targeted therapies and the application of immunotherapy, such as the great achievements in the use of programmed cell death protein-1 (PD-1) and programmed cell death legand-1 (PD-L1) over the past several years, the incidence and mortality have ranked first among all malignant tumors, and the prognosis is still poor [2]. Thus, a further study of the prognosis-relevant factors is warranted.

With the aging of the global population, the number of diabetic patients with lung cancer is increasing each year. It is evaluated that there were approximately 451 million people with diabetes worldwide in 2017 and approximately 8.7% NSCLC patients had type 2 diabetes mellitus (T2DM) [5,6]. T2DM is becoming increasingly prevalent as a metabolic disorder giving rise to a series of serious complications [7,8]. T2DM is reported to be associated with an increased risk of developing several cancers, including colorectal, breast, gallbladder, endometrial, liver, and pancreatic cancer [9]. The status of hyperglycemia and hyperinsulinemia has been considered as a leading proposed mechanism underlying the T2DM-cancer association [10]. Current studies found that insulin and IGF-1 activate pathways stimulated tumor cell proliferation, metastasis, and progression, such as P13K/Akt kinase and Ras/MAP kinase. Hyperinsulinemia also produces peroxide leading to the oxidative stress, which can increase damage to the body caused by mutations in tumor cell-related genes. In addition, some previous epidemiologic studies showed that metformin can reduce the risk and mortality of NSCLC [11]. However, these potential mechanisms have not been fully identified.

Recent researches have implied that comorbidities represent an important contributing factor to the poor prognosis observed in NSCLC patients [5]. As one of the most common chronic disorders, diabetes is considered to have potentially complicated interactions with NSCLC. Large numbers of relevant studies have been published, but inconsistent conclusions have been drawn. A meta-analysis suggested that T2DM has no significant effect on the incidence of lung cancer in men but has a deleterious effect on women [12]. Another meta-analysis revealed that preexisting diabetes has a significantly negative impact on the overall survival (OS) of NSCLC patients [13]. Optimal management of the coexisting diseases can help to improve significantly outcomes [5]. Therefore, we conducted this observational study to evaluate the impact of T2DM on the prognosis of NSCLC patients.

## 2. Methods

### 2.1. Population

This was a retrospective observational patient cohort study to explore the long-term predictors of survival in NSCLC. The inclusion criteria was (1) primarily diagnosed as NSCLC patients aged 18 years old or older; (2) the stage was earlier than IIIA based on tumor-Node-Metastasis (TNM)-based staging; and (3) the pathological type was non-small-cell. The inclusion criteria did not have other restrictions, such as patients’ status or specific type of medical treatments, but data were carefully recorded during the whole enrollment period and their impact on any examined association was explored through rigorous analysis. From January 2019 to January 2020, a total of 247 patients with primary NSCLC were included. All patients were staged according to the guidelines after completing systemic assessment [14]. All procedures followed complied with the ethical standards of the responsible committees on human experimentation (institutional and national). The study protocol was approved by the Ethics Committee of Shanghai Tenth People’s Hospital, Shanghai Tongji University School of Medicine (SHSY-IEC-4.1/20-21/01).

### 2.2. Laboratory Tests

We obtained blood samples after fasting for at least 8 h. The blood samples for fasting plasma glucose (FPG) test were centrifuged on spot in no more than 1 h after collection. Other samples were stored at −20 °C and transported to a central laboratory that is certified by the College of American Pathologists within 2–4 h of collection. FPG was measured by Beckman Coulter AU 680 (Krefeld, Germany). Carcinoembryonic antigen (CEA) was tested by chemiluminescence method (Abbott i2000 SR, Chicago, IL, USA).

### 2.3. Physical Examination and Questionnaire

The medical staff collected information about the demographic characteristics, medical history and personal lifestyle risk factors including smoking, alcohol, and physical activity, by a questionnaire. Every participant was required to be measured by medical staff for height, weight, and blood pressure. Body Mass Index (BMI) was calculated as BMI = weight (kg)/height^2^ (m^2^) as the assessment of obesity. The pathological type, stage and treatments for NSCLC were confirmed from medical records.

### 2.4. Type 2 Diabetes

Diagnosed T2DM was defined as self-reported diabetes previously diagnosed by professional healthcare, which was confirmed by a validated supplementary questionnaire regarding blood tests, duration, and cure. The patients who were found with glycosylated hemoglobin (HbA1c) ≥ 6.5% the first time through the serological test in this study were defined as newly-diagnosed T2DM, according to the American Diabetes Association criteria. Non-diabetes was defined as no history of diabetes as well as HbA1c ≤ 6.4%.

### 2.5. Follow-Up

The study primarily determined the progression-free survival (PFS), which was calculated from the date of the baseline questionnaire to the date of the first occurrence of disease progression or death from any cause. We determined the OS as the date of the baseline questionnaire to the death from any cause. We assessed the long-term survival of patients every 3 months. Follow-up was initiated from the date of enrollment until 1 July 2022 or until patient death. Data were acquired from at least one of the following sources: (1) the hospital medical records; (2) contacting patients or their family by telephone; and (3) the local death registration system. One patient’s medical records were lost. Seven patients had too serious complications, even threatening to their life, to be enrolled in the study. A small proportion (4.5%, *n* = 11) were excluded from the study due to loss of follow-up. Finally, 228 patients with NSCLC were included, as shown in Figure 1.

### 2.6. Data Analysis

Continuous variables were expressed as means and standard deviations. Categorical variables were summarized as percentages and frequencies. All the data were analyzed by the Statistical Package for Social Sciences (SPSS), version 24.0. We used ANOVA to test the differences among groups. Pearson χ^2^ test was used for the comparation of categorical variables. Survival curve was used to examine the difference of progression-free survival. We used multinomial logistic regression to assess the association of T2DM with prognosis of lung cancer. The hazard ratios (HR) and 95% confidence intervals (CI) for death were calculated by using the Cox proportional hazards regression model. *p* < 0.05 was considered as statistically significant.

## 3. Results

### 3.1. General Characteristics

The study enrolled 144 men and 84 women. All patients were followed-up for more than 24 months. Basic patient and tumor characteristics are summarized in Table 1. In total, 150 patients had a history of smoking, a third had already quit. A total of 128 patients had early stage lung cancer, while 100 patients were stage III. A total of 69.8% of patients had adenocarcinoma. All adenocarcinoma patients underwent genetic testing, 75 of who had EGFR mutation, and 3 had ALK mutation. All lung cancer patients were treated according to the patient’s condition and stage, including operation, chemotherapy, targeted therapy, immunity therapy, and radiotherapy based on the ASCO guidelines for lung cancer and personal preference. The overall percentage of type 2 diabetes was 38.2%, and 12 of them were newly-diagnosed as type 2 diabetes after lung cancer; 45 patients took oral medication, while 21 patients subcutaneously injected insulin, and 15 patients took both oral medication and insulin as treatment for type 2 diabetes.

### 3.2. Comparison of Prognosis of Lung Cancer in T2DM Groups

To compare the clinical characteristics in different type 2 diabetes statuses, all the participants were categorized into three subgroups categorized by diagnosis of type 2 diabetes, as shown in Table 2. NSCLC patients with type 2 diabetes were relatively older, especially the newly-diagnosed ones. Fasting glucose and HbA1c of diabetes patients were much higher. There were no significant differences in the baseline characteristics of tumors between the diabetes group and the non-diabetes group, including tumor size, pathological type proportion, gene mutation proportion, and treatment. However, OS differed among the groups. Both of the OS and PFS of lung cancer patients with diabetes were longer than the other two groups, and were the shortest for patients without diabetes. The survival curve in Figure 2 intuitively shows that the OS and PFS of the patients without diabetes was shorter than for those with diabetes.

### 3.3. Association of T2DM with Prognosis of Lung Cancer

As shown in Table 3, compared with the non-diabetes group, diagnosed T2DM was associated with the prognosis of lung cancer after adjusting for age and covariates. T2DM patients tended to have longer OS. In addition, taking metformin was associated with the OS of lung cancer. After the adjustment for age and covariates, the relation tended to lose statistical significance. Insulin treatment seemed to have no association with the prognosis of NSCLC. Figure 3a has shown the factors affecting OS in all patients intuitively, which were diabetes, staging of tumor and initial CEA. We found that metformin had a slight protective influence on OS from Figure 3b. The association between T2DM and OS was influenced by age, stage and cancer treatment. Figure 3c has demonstrated the associations of OS with status of T2DM in age group. T2DM was associated with OS only in patient aging between 60–70 years old. Among the different stages in Figure 3d was associated positively with OS in early stage of NSCLC, not in the middle stage. Compared with immunotherapy and targeted therapy, T2DM was associated with OS after operation and chemotherapy as shown in Figure 3e.

## 4. Discussion

T2DM is nowadays becoming a major global health problem, resulting in significant morbidity and mortality. The prevalence of T2DM in China is high to 9.1% causing huge socio-economic burdens [15]. T2DM is an independent risk factor in the development of NSCLC [16], and has also been reported as a negative impact on the short-term outcomes of patients with surgically treated NSCLC [17,18]. A meta-analysis suggested pre-existing T2DM may increase the risk of lung cancer, especially among women with diabetes [19]. However, evidence on the correlation between T2DM and prognosis of NSCLC is still indefinite and conflicting, requiring more researches. To derive relatively objective evidence of the prognostic value of T2DM in NSCLC, we performed this observational study.

In our study, we found that the OS and PFS of NSCLC patients with diabetes were longer than for the non-diabetes group. The status of diagnosed T2DM was associated with the prognosis of NSCLC in early and middle stage after adjustment age and covariates. The patients with both NSCLC and T2DM tended to have longer PFS and OS. The association between T2DM and OS was influenced by age, stage and cancer treatment. T2DM had a slight protective influence on OS in early stage cancer patients aging from 60 to 70 years old after operation or chemotherapy. Since our results seemed differ from most current research, the impact of diabetes on the prognosis of lung cancer is still controversial. A large prospective study provided evidence that pre-existing diabetes is associated with lower OS among female patients with NSCLC [20]. Another meta-analysis indicated that T2DM is significantly associated with a shorter OS in lung cancer patients, but no significant association between T2DM and the OS in NSCLC patients [6]. A current study found that patients with DM had significantly worse OS than those without after surgery of NSCLC, but no strong evidence of a significant difference in PFS [21]. The genuine biological and pathophysiological linkage between DM and NSCLC is now uncertain. Hyperglycemia and hyperinsulinemia following a series of metabolic disorder may be the potential factors of the development of NSCLC [22,23]. The metabolism of tumor cells is under anaerobic conditions, so the hyperglycemia provides better survival conditions for tumor cells. Some studies have reported that T2DM, characterized by insulin resistance, resulted to chronic hyperinsulinemia enhancing growth hormone receptor expression and increasing IGF-1 receptor production, eventually activating IGF-1 receptor signaling pathway, which was known as a promoter of tumor progression [24,25]. EGF pathways are also regarded with cancer metabolism. The inhibition of EGF pathways may have a synergistic antitumor effect in NSCLC through the reactivation of oxidative phosphorylation [26]. To some degree, these results may explain the reason why NSCLC patients with T2DM have a significantly worse prognosis.

Our study found T2DM an independent favorable prognostic factor for NSCLC, seemingly in contrast to most published literature. We also found that taking metformin was correlated with the OS of lung cancer with weak statistical significance, which might explain the different results. The effect of T2DM on the prognosis of NSCLC patients may be influenced by the administration of antidiabetic therapeutics, such as metformin. In recent years, a fair amount of evidence has shown that metformin can reduce the mortality of lung cancer. The mechanisms mainly can be improving hyperinsulinemia and insulin resistance leading to activating adenosine monophosphate-activated protein kinase pathway. Metformin can also promote tumor apoptosis and inhibit inflammatory response [4,27,28]. Metformin has been proven by some researches to inhibit cell proliferation, induce apoptosis, and block cell cycle progression in NSCLC cells via LKB-1/AMPK signaling pathway [29]. Additionally, Luo et al. found that lung cancer cells could be destroyed by metformin via AMPK/PKA/GSK-3β-axis-mediated surviving degradation [30]. Besides basic research, some clinical studies have reported the relationship between metformin and the prognosis of lung cancer. A current meta-analysis found that use of metformin was not associated with OS of patients with lung cancer [31]. In another study by Menamin et al., metformin use after diagnosis showed more beneficial effects on lung cancer-related mortality than use before diagnosis [32]. In a recent population-based cohort study, metformin use was corelated with increased survival of lung cancer patients [33]. Despite of the amount of research, the correlation between metformin and the prognosis of NSCLC is still controversial. In addition, another explanation for our results is that T2DM patients may pay more attention to their own health. For example, T2DM patients may be examined more frequently and have more chance of being diagnosed at early stages. However, this proposed hypothesis was not supported by the current data showing no differences in tumor-related characteristics with and without diabetes.

Some limitations of our study should be considered. First, the amount of 247 patients was not large enough, which might lead to deviation of statistical results. Second, even though we took several covariates into account, there were still some other confounding variables being missed, such as family history, etc. Finally, our study being a retrospective cohort study might reduce the statistical power of our results. Therefore, further studies are required to reveal the actual interaction of T2DM with NSCLC.

## 5. Conclusions

In conclusion, the OS and PFS of NSCLC patients with diabetes were longer than for those without diabetes. T2DM is an independent prognostic factor for NSCLC staging earlier than IIIA. The association between T2DM and OS was influenced by age, stage and cancer treatment. T2DM had a slight protective influence on OS in early stage cancer patients aging from 60 to 70 years old after operation or chemotherapy. The patients with both NSCLC and T2DM tended to have a longer OS, possibly due to metformin. Our study may provide a new perspective on the prognostic factors of lung cancer and provide certain evidence for subsequent adjuvant therapy of lung cancer. According to our results, metformin may benefit lung cancer prognosis. In any case, further research should be performed to determine the causal relationship and to explore the underlying mechanisms.

## Figures and Tables

**Figure 1 jcm-12-00321-f001:**
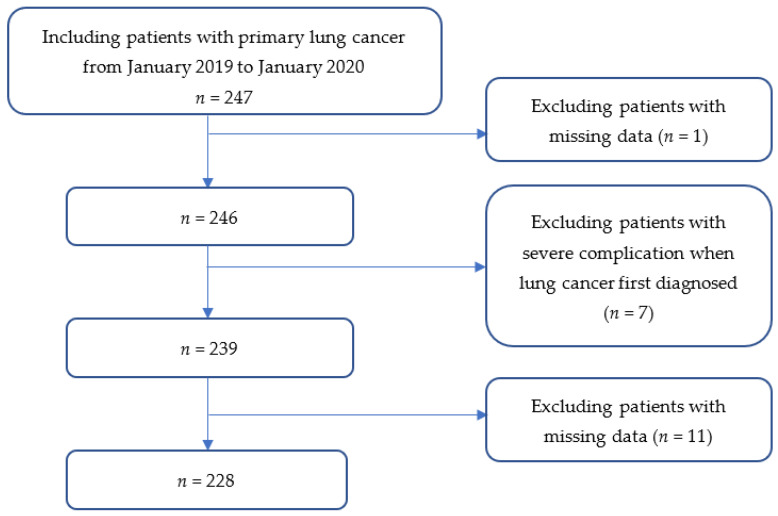
Enrollment and follow-up of the patients until 1 July 2022 or until patient death.

**Figure 2 jcm-12-00321-f002:**
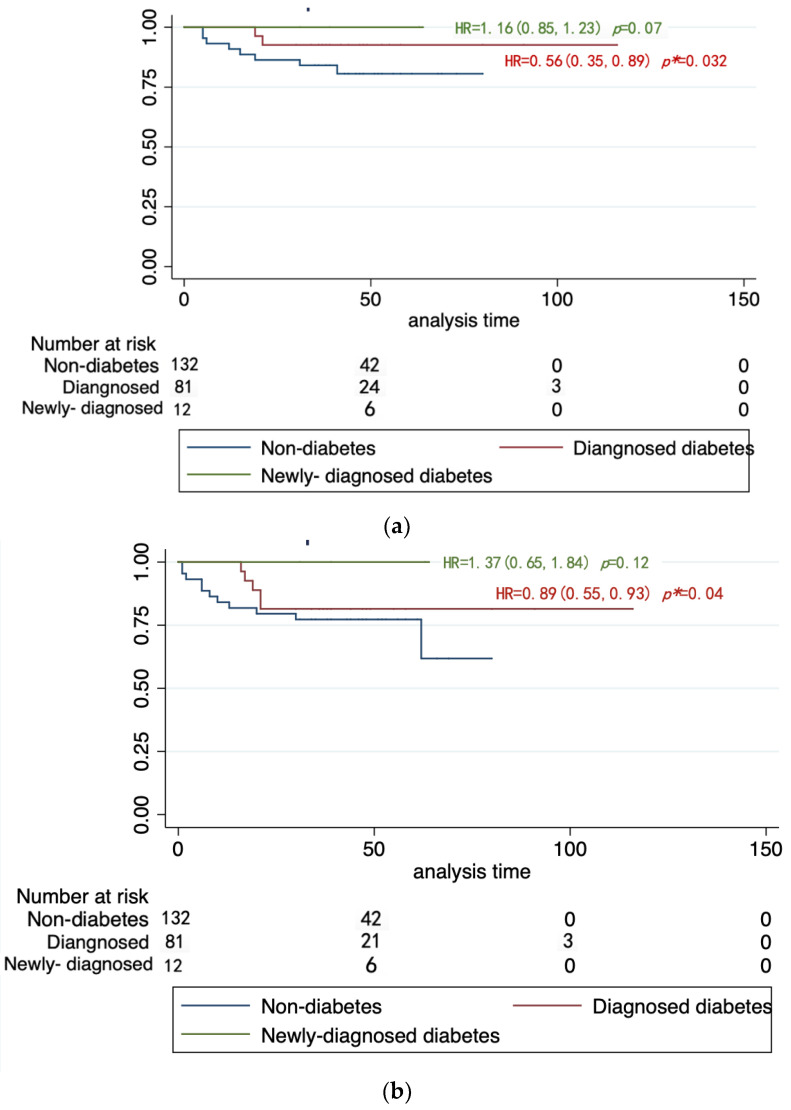
(**a**). OS survival curves of NSCLC in diabetes groups. (**b**). PFS survival curves of NSCLC in diabetes groups. * indicates significance *p* < 0.05.

**Figure 3 jcm-12-00321-f003:**
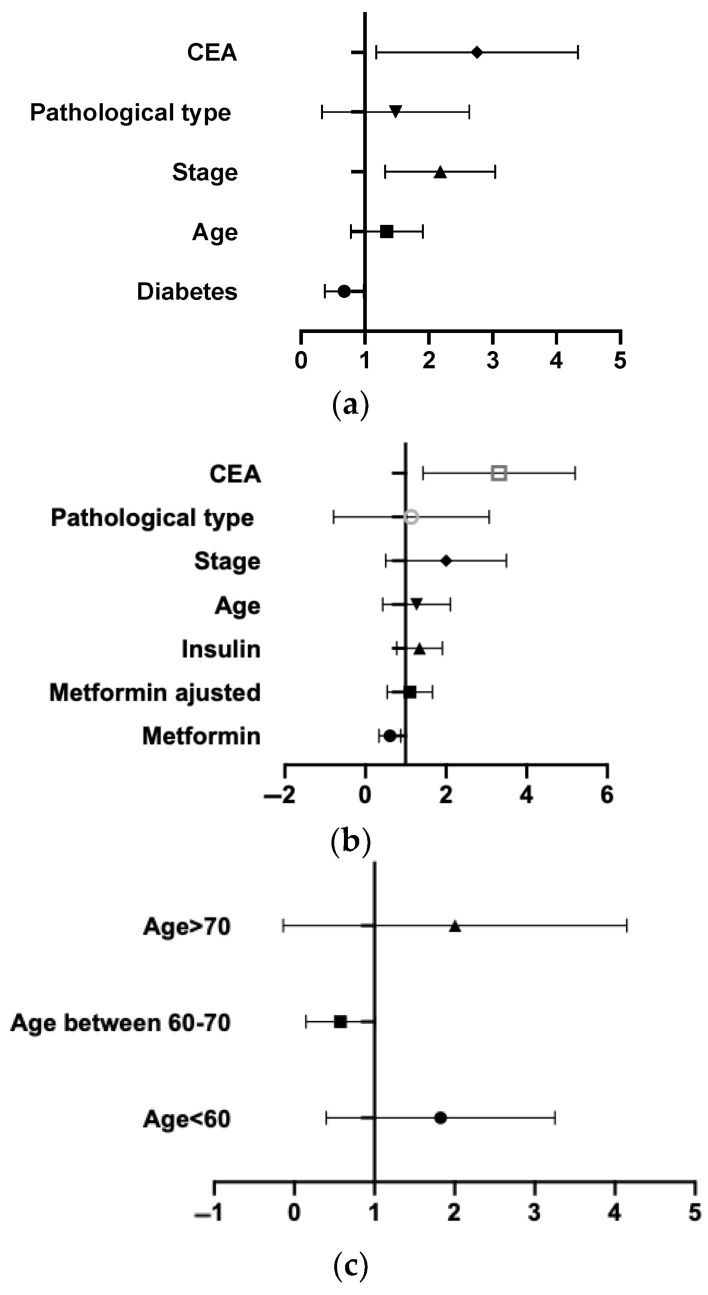
Associations of OS with status of T2DM in different groups. All the five figures were hazard ratio (95% confidence interval). (**a**) showed the impacts affecting OS in all patients, including age, stage, pathological type, CEA etc. (**b**) showed the impacts affecting OS in diabetes patients, including diabetic therapy, age, stage, pathological type, CEA etc. (**c**) showed the associations of OS with status of T2DM in age group. (**d**) showed the associations of OS with status of T2DM in different stage. (**e**) showed the associations of OS with status of T2DM in different cancer treatment.

**Table 1 jcm-12-00321-t001:** Characteristics of participants.

Characteristics	Participants
Age (mean, SD)	63.77 ± 8.34
Gender (*n*, %)	
Male	144 (63.2%)
Female	84 (36.8%)
Smoking history (*n*, %)	
Smoker	99 (43.3%)
Quit smoker	51 (22.4%)
Non-smoker	78 (34.2%)
Pathological type (*n*, %)	
Squamous cell carcinoma	66 (28.9%)
Adenocarcinoma	159 (69.8%)
Neuroendocrine carcinoma	3 (1.3%)
Gene mutation (*n*, %)	
EGFR	75 (32.9%)
ALK	3 (1.3%)
Non-mutation	150 (65.8%)
Stage (*n*, %)	
IA-IIB	128 (56.1%)
IIIA-IIIC	100 (43.9%)
Therapy (*n*, %)	
Operation	144 (63.2%)
Chemotherapy	207 (90.8%)
Targeted therapy	66 (28.9%)
Immunity therapy	69 (30.3%)
Radiotherapy	15 (6.6%)
Diabetes (*n*, %)	
Diabetes	75 (32.9%)
Newly-diagnosed diabetes	12 (5.3%)
Non-diabetes	141 (61.8%)
Treatment to diabetes (*n*, %)	
Diet control only	6 (6.9%)
Oral medication	45 (51.72%)
Insulin	21 (24.14%)
Oral medication + insulin	15 (17.24%)

EGFR, epidermal growth factor receptor; ALK, anaplastic lymphoma kinase.

**Table 2 jcm-12-00321-t002:** Comparison of tumor related characteristics in different status groups of T2DM.

	Non-Diabetes (*n* = 141)	Newly-Diagnosed Diabetes (*n* = 12)	Diagnosed Diabetes (*n* = 75)	*p*
Age (years)	61.74 (9.14)	67.25 (2.87)	66.63 (6.34) ^#^	0.037 *
Duration (months)	/	14.5 (5)	102.3 (82.43)	0.004 *
BMI (kg/m^2^)	23.37 (3.41)	25.27 (2.60)	23.87 (3.08)	0.56
FPG (mmol/L)	5.49 (0.64)	7.62 (0.92) ^#^	7.74 (2.07) ^#^	0.000 *
HbA1c (%)	5.98 (0.61)	7.48 (1.02) ^#^	7.75 (1.88) ^#^	0.00 *
Smoking history (%)	53.19%	75%	77.78%	0.181
Tumor size (cm)	2.61 (2.05)	2.07 (0.75)	2.7 (1.34)	0.847
CEA	22.64 (48.02)	5.33 (1.24)	11.56 (13.48) ^#^	0.04 *
Pathological type (%)				0.769
Squamous cell carcinoma	32.91%	7.7% ^#^	24%	
Adenocarcinoma	65.96%	38.46%	61.29%	
Staging (%)				0.23
IA-IIB	55.32%	50%	60%	
IIIA-IIIC	44.68%	50%	40%	
Gene mutation (%)	36.17%	25%	32%	0.510
Operation (%)	59.57%	75%	68%	0.894
Chemotherapy (%)	87.36%	75%	96%	0.209
Targeted therapy (%)	17.02%	25%	8%	0.312
OS (months)	41.39 (16.83)	49.25 (16.78) ^#^	47.74 (20.48) ^#^	0.031 *
PFS (months)	39.39 (20.61)	49.25 (16.78) ^#^	44.93 (22.50) ^#^	0.027 *

Values are mean (SD) unless otherwise noted. * Significantly different among all the three group (*p* < 0.05). ^#^ Significantly different between non-diabetes group and individual group (*p* < 0.05). BMI, body mass index; FPG, fasting plasma glucose; HbA1c, glycosylated hemoglobin; CEA, carcinoembryonic antigen; EGFR, epidermal growth factor receptor; ALK, anaplastic lymphoma kinase.

**Table 3 jcm-12-00321-t003:** Associations of OS with status of T2DM and with diabetic therapy.

	HR (95% CI) for OS	*p*
Diabetes status		
Non-diabetes	1.00 (ref.)	
Diabetes	0.67 (0.38, 0.99) *	0.047 *
Diabetes ^#^	0.80 (0.43, 0.97) *	0.046 *
Diabetes ^##^	0.88(0.44, 1.24)	0.31
Diabetic therapy		
Untreated	1.00 (ref.)	
Metformin	0.53 (0.39, 0.91) *	0.043 *
Metformin ^#^	0.72 (0.52, 0.98) *	0.046 *
Metformin ^##^	0.82 (0.74, 1.75)	0.086
Insulin	1.26 (0.83, 1.95)	0.844
Insulin ^#^	1.22 (0.79, 1.75)	0.869
Insulin ^##^	1.28 (0.78, 2.67)	0.862

Multinomial logistic regression analyses were performed. Data were odds ratio (95% confidence interval). * indicates significance *p* < 0.05. ^#^ Adjusted for age. ^##^ Adjusted for age and covariates including DM duration, CEA, pathological type and staging. Ref., reference.

## Data Availability

The data presented in this study are available on request from the corresponding author. The data are not publicly available due to privacy and ethical restrictions.

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
