# Peer review of "Impact of Type 2 Diabetes Mellitus on the Prognosis of Non-Small Cell Lung Cancer"

_jcm, 2022, doi:10.3390/jcm12010321_

Round 1

Reviewer 1 Report

The current manuscript titled: "Impact of Type 2 Diabetes Mellitus on the Prognosis of Non-small Cell Lung Cancer" represents an important analysis of evolving field of Oncology.

The title reflects the manuscript content and helps the reader navigate the article essence.

In my opinion, these are the adjustments which should be made to increase the value of your manuscript:

1.      Line 10: please, abbreviation for “NSCLC”.

2.      Line 11: please, abbreviation for “OS” and “PFS”.

3.      In Introduction chapter, please, add more detailed information about Non-small Cell Lung Cancer. Moreover, please add information about the pathophysiological processes linking Non-small Cell Lung Cancer and Type 2 Diabetes Mellitus.

4.      In the Discussion section, there is not enough comparative information with other studies.

5.       In the Conclusions section, add and highlight the practical utility and significance of this study.

6.       The manuscript contains some punctuation errors, please revise the text (e.g., lines 197, 200, 202, etc.).

Author Response

Please see the attachment. Thank you for your review. 

Reviewer 2 Report

Many studies have demonstrated metformin can inhibit cancer cells proliferation not only on lung cancer but also various cancers such as breast cancers, colorectal cancer, pancreatic cancer, and so on. Therefore, the result of this study is not important and novel information. In order to provide a complete data, the Author may further analyze the OS and PFS of lung cancer patients with metformin treatment and non- metformin treatment groups before published. 

Author Response

(The authors gave the same response as above.)

Reviewer 3 Report

The authors conducted an interesting study on patients suffering from lung cancer and analyzed the possible correlation with diabetic pathology.

However, the presentation of the study needs further improvement.

Major queries:

a) In Table 2 add p-values between individual groups.

b) In Figure 2 add at-risk patients (at the bottom of the figures) for individual groups; the p-values between survival curves; the confidence intervals at three (at least) time points on the curves.

c) Table 3:  How have the covariates been chosen?

Minor queries:

a) Add the authors' affiliations.

b) In Section 2.3 correct the BMI formula.

c) In the Figure 3 caption the figures are five, not four as stated.

d) At the end of Section 3.3, a reference to Figure 3(e) is missing.

e) Move Table 2 before Figure 2, as the former is mentioned earlier in the manuscript.

f) The caption of Table 3 is a bit confusing, it needs some adjustments.

g) In Table 3 the numeric values need to be rounded at the same number of decimal digits (two are enough).

Author Response

Please see the attachment. Thank you for the review. 

Round 2

Reviewer 1 Report

I agree with the changes made, which significantly improve the quality of the manuscript. I recommend this article for publication.

Reviewer 2 Report

The sentence may be improved